# Photophore Morphogenesis and Extraocular Encephalopsin Expression during the Embryogenesis of Smalleye Pygmy Shark (*Squaliolus aliae*)

**Laurent Duchatelet [1,*], Hsuan-Ching Ho [2,3] and Jérôme Mallefet [1]**

1 Marine Biology Laboratory, Earth and Life Institute, Université catholique de Louvain, 1348 Louvain, Belgium
2 National Museum of Marine Biology and Aquarium, Pingtung 94450, Taiwan
3 Department and Graduate Institute of Aquaculture, National Kaohsiung University of Science and Technology, Kaohsiung 80778, Taiwan
* Correspondence: laurent.duchatelet@uclouvain.be

**Abstract:** Bioluminescence is a common phenomenon in marine organisms, especially in deep water where faint blue light remains. Among elasmobranchs, three families display the ability to emit light, the Etmopteridae, the Dalatiidae, and the Somniosidae. Luminous sharks have thousands of minute light organs, called photophores, that are mainly present ventrally and produce light. The main function of shark luminescence is counterillumination to camouflage the shark silhouette by mimicking the residual ambient light and avoiding being spotted by predators underneath. To perform counterillumination efficiently, luminescence needs to be finely adjusted. A new type of control was recently demonstrated via extraocular photoreception at the level of the light organ. An encephalopsin (i.e., opsin 3) was shown to be expressed in the vicinity of the photophore of an Etmopteridae species, *Etmopterus spinax*. This opsin was also demonstrated to be expressed concomitantly with the photophore development (i.e., when photophores become able to produce light) during *E. spinax* embryogenesis. To understand the photophore morphogenesis of different shark families, we analyzed the smalleye pygmy shark, *Squaliolus aliae*, with a photophore formation which represents the first report on the Dalatiidae family. Since Dalatiidae and Etmopteridae are phylogenetically closely related, the photophore morphogenesis was compared with an Etmopteridae representative, *Etmopterus spinax*. The results also reveal that *Squaliolus aliae* shares similar encephalopsin expression pattern as in *Etmopterus spinax*, which further supports evolutionary conservation of photophore morphogenesis as well as its own encephalopsin-based light perception across the two luminous shark families.

**Keywords:** bioluminescence; extraocular opsin; luminous shark; Dalatiidae; morphogenesis evolutive conservation

## 1. Introduction

Bioluminescence plays an important ecological role for marine organisms [1,2]; a large set of functions is associated with the photoemission phenomenon [1,3,4]. Emission of light may aid in predation, predation avoidance, or intraspecific communication [1]. To achieve these ecological functions efficiently, bioluminescence needs to be finely regulated. Most marine organisms use nervous compounds (i.e., neurotransmitters) to control light emissions; for example, cnidarian, ctenophores, and most fishes control light production via adrenergic stimulation [5–10], and echinoderms trigger luminescence through a cholinergic control [11–13]. Neuromodulators such as nitric oxide may also play a role in neural-induced luminescence, such as for the krill, *Meganyctiphanes norvegica*, and some fishes [14–16]. Recently, another form of hormonal physiological control has been exclusively found in luminescent elasmobranchs [17–19]. Light emission is triggered by melatonin, while inhibited through melanocortin action. Melatonin and melanocortin

receptors were shown to be expressed within the light organ [20]. Recently, literature reports increased evidence of a duality between photoemission and photo-perception in bioluminescent organisms. From reports of opsin expression within luminous cells in ctenophore to functional coupling mechanisms regulating light output in lanternshark, the link between the two photobiological processes is suggested to play a role in luminescence regulation in various marine organisms [21–25].

Among elasmobranchs, three shark families are known to produce light: the Dalatiidae, the Etmopteridae, and the Somniosidae [19]. Recent Squaliformes' phylogenetic analyses put the origin of shark luminescence at the root between Dalatiidae and the two other families [19]. These sharks emit light via thousands of minute light organs (i.e., photophores) spread mainly in the ventral epidermis. Ultrastructurally depicted, etmopterid species display photophores composed of a cup-shaped pigmented sheet upholstered by a guanine crystal reflective layer encasing several light-producing cells (i.e., photocytes) [26–28]. These structures are surmounted by melanophore-like cells, forming the iris-like structures (i.e., ILS), acting as a light organ shutter by pigment motion. Finally, one or two lens cells are located above the ILS focusing the light produced to the outside [28]. Photocytes contain three different cytological parts: the nucleus, the basal granular area, and the apical vesicular area, which contain fluorescence vesicles, named glowons, assumed to be the site of light production [29]. Morphogenesis of etmopterid photophores during shark embryogenesis starts with an agglomeration of pigmented cells, followed by the apparition of protophotocyte (i.e., pre-photogenic cells without auto-fluorescent vesicle and enable to produce light) and associated ILS and lens cells [30]. During the final embryonic stage, photophores display matured photocytes and produce luminescence [30]. Interestingly, encephalopsin (i.e., opsin 3) expression was demonstrated to appear in the ILS cell membranes during this final stage, suggesting early photo-perception within the photophore [31]. The classical histology of adult dalatiids (*Squaliolus aliae*, *S. laticaudus*, *Isistius brasiliensis*, and *Dalatias licha*) and somniosid (*Zameus squamulosus*) photophores has been depicted previously. Both photophore types are less complex than *Etmopterus spinax*, displaying a unique photocyte, embedded between pigmented cells and surmounted by lens cells [19,32–36], and a less dense ILS is also present. Somniosidae present atypical bioluminescent squamation with overlapping honeycomb placoid scales translucent to light, while dalatiids generally have pavement-shape placoid scales [19,37].

The smalleye pygmy shark is one of the smallest sharks (maximum adult size of 24.1 cm) living in the southeastern Indian and west Pacific Ocean pelagic zones from the surface to 2000 m [38,39]. To date, most of the ecology (e.g., lifespan, litter size, reproduction strategy) of this species remains unknown. Even if the diet of this midwater ovoviviparous shark remains enigmatic, *Squaliolus laticaudus*, a phylogenetically closed species mainly feeds on small midwater fishes, cephalopods, and crustaceans [40]. This work aims to investigate the photophore's and associated structures' morphogenesis, with a specific emphasis on the apparition of a homologous encephalopsin, during the embryonic development of a dalatiid species, *Squaliolus aliae* (Figure 1). Similar morphogenesis as an etmopterid and an association between light production and photo-perception through an encephalopsin-like is found, which suggests a conservative evolution of control mechanism among luminous sharks.

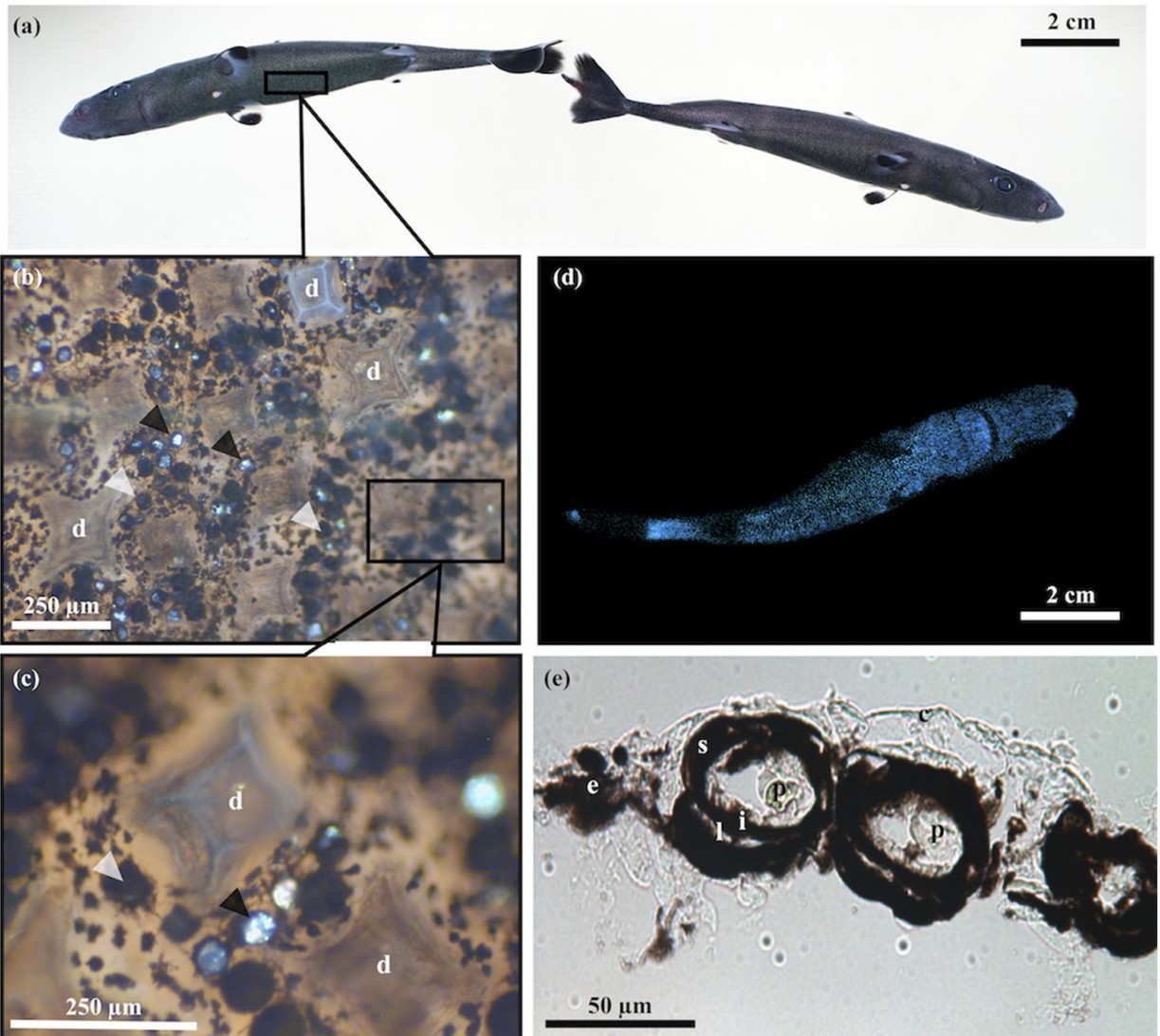

**Figure 1.** General aspects of *Squaliolus aliae* luminescent structures. (**a**) Ventral (left) and lateral (right) pictures of adults *S. aliae*. (**b**) Placoid scales and photophore observation on a ventral skin patch. (**c**) Closer view of typical pavement-shaped placoid scales surrounded by open and closed photophores. (**d**) Picture of the ventral side luminescent pattern of in utero *S. aliae* embryo. (**e**) Cross section of the photogenic skin of adult *S. aliae* showing classical photophore histology. Dark arrowhead: closed photophore; white arrowhead: open photophores. d: dermal scales; e: epidermis; i: iris-like structure; l: lens cell; p: photocyte; s: pigmented sheath.

## 2. Materials and Methods

### 2.1. Specimen Collection

Sixteen gravid female smalleye pygmy sharks, *S. aliae* (19.28 ± 2.49 cm TL), were captured as bycatch during midwater trawls (50 to 150 m depth) in inshore waters off Donggang harbor in southwest Taiwan (22°26′ N, 120°23′ E). All specimens were captured during two field seasons (July 2011 and December 2018). Specimens were brought to the National Museum of Marine Biology and Aquarium (Pingtung, Southern Taiwan), sized, weighed, and female uteri were excised to free embryos. When possible, embryos' and adults' luminescence was pictured in dark conditions using Sony α7SII camera. A total of sixty embryos (1.7 to 11.7 cm TL) were extracted from sixteen different litters.

Embryos and females were fixed in phosphate buffer saline with 4% paraformaldehyde overnight before being stored in fresh phosphate buffer saline until use. Three eyes were also dissected, fixed, and stored to perform controls.

## 2.2. Photophore Morphology and Development

All embryos and females were measured and photographed with a digital camera (Panasonic DMC-FZ300, Panasonic, Kadoma-shi, Osaka, Japan) to establish a developmental stages classification, considering the embryo size as a proxy of their relative age (larger specimens are supposed to be older). Ventral skin patches of 1 cm$^2$, located between pectoral fins, were dissected for each embryo and photographed under a stereomicroscope (Leitz Diaplan, Wild Leitz, Wetzlar, Germany) coupled with a digital camera (ToupTek UCMOS 03100 KPA, Hangzhou ToupTek Photonics, Hangzhou, Zhejiang, China). Pictures were used to determine the photophore and placoid scale density, the mean scale sizes, and skin pigmentation status using ImageJ software (ImageJ, National Institutes of Health, Bethesda, Rockville, MD, USA, https://imagej.nih.gov/ij/, accessed on 6 November 2022, 1997–2018). To evaluate densities, three pictures per specimen, covering a skin area of 0.681 mm$^2$ each, were randomly taken. Photophores and scales present in the image were counted, and an average density was made per individual. The density was transposed into the number of photophores per cm$^2$ and the number of placoid scales per cm$^2$. The scale size corresponding to the maximum width of the pavement-like structure, which was measured on each scale in the same area. For both density and scale size, measurements were taken on at least five specimens per developmental stage. The skin pigmentation status was established based on presence/absence.

## 2.3. Encephalopsin Immunodetection

Skin patches and eye retinas were cryoprotected through a series of PBS baths with increasing sucrose concentrations (10% for 1 h, 20% for 1 h, and 30% overnight). Then, tissues were embedded with an optimal cutting medium (OCT compound, Tissue-Tek, Sakura Finetek, Torrance, CA, USA) and immediately frozen at $-80$ °C. Sections of 10 μm were obtained with a cryostat microtome (CM3050 S, Leica Microsystems GmbH, Wetzlar, Germany) and laid on coated slides (Superfrost slide, Thermo Scientific, Waltham, MA, USA). To detect and localize encephalopsin, the immunofluorescence technique was used, following [31], on *S. aliae* adult and embryo skin and retina sections. Slides were rinsed in Tris buffer saline 1% Tween (TTBS: Trizma base (Sigma) 20 mM, NaCl 150 mM, pH 7.5 + 1% Tween 20 (Sigma)), then blocked with TTBS containing 10% bovine albumin serum (BSA, Amresco, Cleveland, OH, USA). Slides were incubated overnight with the anti-encephalopsin primary antibody (anti-encephalopsin Pab in *Homo sapiens*, Genetex, GTX 70609, lot number 821,400,929) at a dilution of 1/400 in TTBS 5% BSA. Slides were bathed again in TTBS for 30 min before being incubated in the dark with the secondary antibody Alexa Fluor® 594 Goat Anti-Rabbit IgG (Goat Anti-Rabbit, Alexa Fluor® 594, Life Technologies Limited, Inchinnan, UK) at a dilution of 1/200 in TTBS 5% BSA and rinsed 15 min in TTBS. Finally, slides were subjected to 15 min DAPI (DAPI nucleic acid stain, Invitrogen, Waltham, MA, USA) nucleus staining, rinsed for 10 min in TTBS, and mounted with Mowiol (Mowiol® 4–88, Sigma, St. Louis, MI, USA). Sections were observed under a Polyvar SC epifluorescence microscope (Leica Reichter Jung) equipped with a Nikon DS-UI digital camera coupled with NIS elements FW software (Nikon Instruments Inc., Melville, NY, USA). Control sections (i.e., omission of the primary antibody as well as immunodetection in the retina) were performed similarly. Encephalopsin immunodetections were performed on at least two embryos of the same established embryonic stage, three females, and applied on at least ten sections for each specimen.

## 2.4. Statistical Analyses

All statistical analyses were performed using RStudio v. 1.2.5019 software (RStudio Inc., Boston, MA, USA) and considered significant for *p*-value < 0.05. Normality and equality of variances were tested by a Shapiro–Wilk test and a Bartlett's test, respectively. As the data do not follow the normal distribution, a non-parametric Kruskal–Wallis ANOVA test was performed to determine whether there are significant differences between the different groups. When the Kruskal–Wallis test indicated that a significant differ-

ence existed between the groups, all pairwise comparisons were tested using a post hoc Dwass–Steele–Critchlow–Fligner comparison test (dscf all pairs test). For linear regression, the normality and homogeneity of the residues were tested by a Shapiro–Wilk test and a Breush–Pagan's test, respectively. For each stage of development (n = 5), corresponding to the number of sharks used, an average of three measurements were made on each specimen to calculate photophore densities and scale densities as well as an average of the sizes made on twenty scales randomly selected per shark specimen.

## 3. Results

### 3.1. Skin Morphometry

*Squaliolus aliae* embryos were classified into six different stages of embryonic development (Stage I to VI), with stage VII being adulthood individuals (Figure 2a). The classification was made according to different criteria: (i) the embryo size, (ii) the placoid scale development, and (iii) the state of skin pigmentation. According to embryo size, specimens were sorted as follows: stage I—less than 4 cm, stage II—from 4 to 6 cm, stage IV—from 6 to 8 cm, stage V—from 8 to 9 cm, and stage VI—over 9 cm. Embryos with a medium size between stage II and IV, but with placoid scale and pigmentation developments intermediate from those stages, were classified as stage III. Placoid scales appear in stage II in the form of fine circular translucent scales (Figure 2b), while in stage IV, specimens present fully formed pavement-shaped scales as observed in adulthood (Figures 1b,c and 2b). Skin pigmentation also appears at stage II and increases during embryonic development. The first observations of protophotophores are performed during stage III. In subsequent stages, the number of photophores increases to reach a mean adult photophore density of $7863 \pm 636$ photophores per cm$^2$. Statistical differences are shown between the different stages in term of photophore density ($\chi^2 = 101.69$, *p*-value < 0.0001), scale density ($\chi^2 = 63.29$, *p*-value < 0.0001), and scale size ($\chi^2 = 323.97$, *p*-value < 0.0001) (Supplemental File S1).

The average scale size increases only slightly, while the scale density remains stable during the shark's growth (Figure 3a). The photophores density, in comparison with scales density, increases significantly during ontogenesis (Figure 3b).

### 3.2. Photophore Histology and Development

The *S. aliae* luminous organ is composed of different parts. In adults, the photophore is delimited by a layer of chromatophore-like cells, called the pigmented sheath, encasing the photocyte, which presents a fluorescent vesicle. The photophore is capped by an ILS, formed by the juxtaposition of pigmented cells, which is surmounted by several lens cells (Figure 1e).

During the development of the photophore, different steps can be distinguished (Figure 2c). Firstly, pigmented cells appear between the epidermis and the connective dermal tissue (Figure 2c—Stage II). Then, pigmented cells develop and form the pigmented sheath (Figure 2c—Stage III). ILS and lens cells appear as well as a non-fluorescent photocyte, called the protophotocyte (Figure 2c—Stage III). The final step of photophore development occurs when fluorescent vesicles appear in the photocyte (Figure 2c,d—Stage IV). At that final stage, *S. aliae* embryos can produce light (Figure 1d).

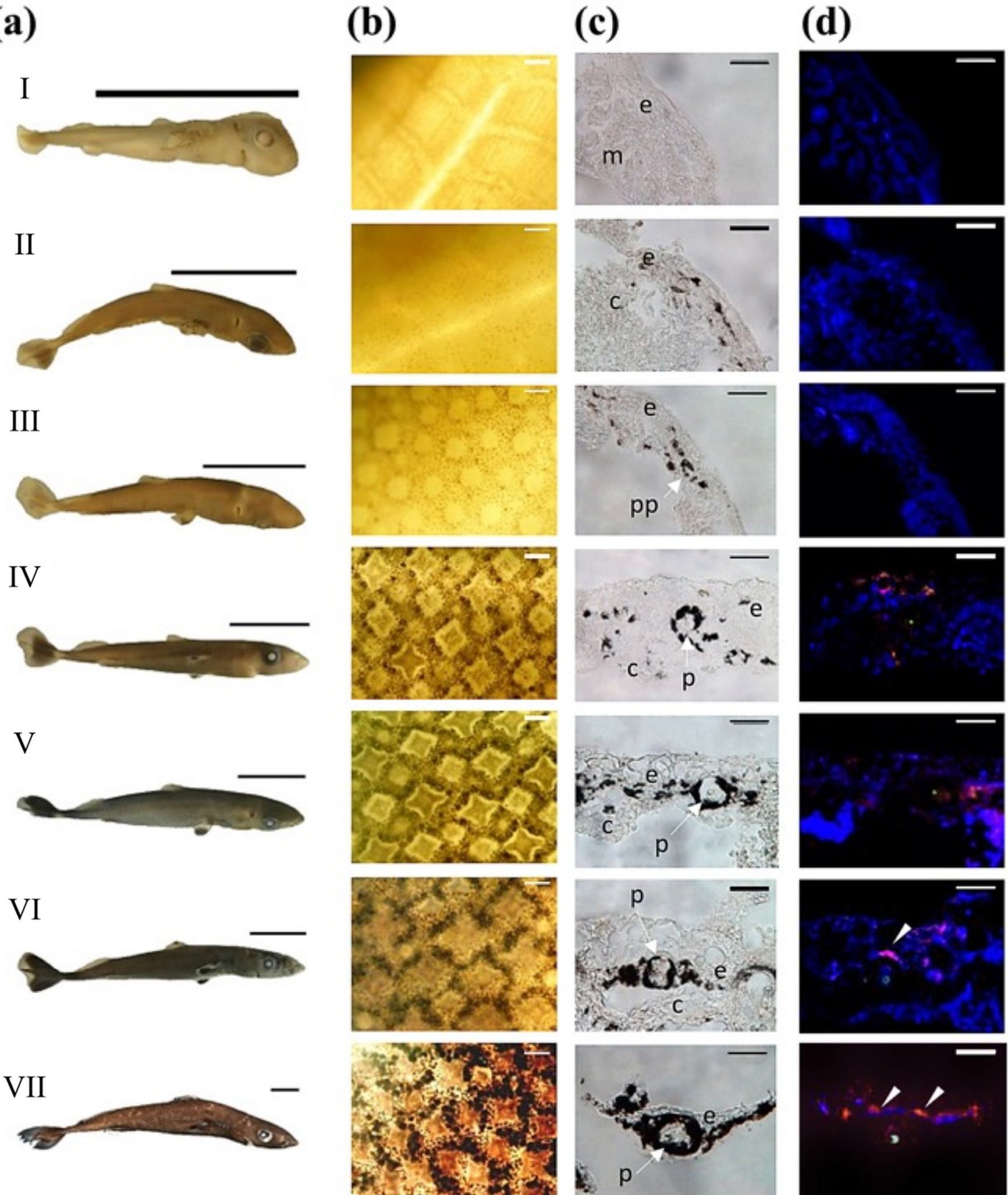

**Figure 2.** Skin pattern, photophore morphogenesis, and encephalopsin labeling during *Squaliolus aliae* embryogenesis. (**a**) Developmental stages (stage I to VI: embryonic stage; stage VII: adult). (**b**) Scales and photophore development on ventral skin. (**c**) Photophore morphogenesis visualization on ventral skin cross sections. (**d**) Encephalopsin immunolabelling during the photophore morphogenesis. Green autofluorescence corresponds to matured vesicles within the photocyte, blue staining corresponds to DAPI nucleus labeling, and red labeling (white arrowhead) corresponds to the encephalopsin immunoreactive site. Scale bars: (**a**) 2 cm, (**b**) 200 μm, and (**c,d**) 100 μm. c: Connective tissue; e: epidermis; m: muscle; pp: protophotophore; p: photophore.

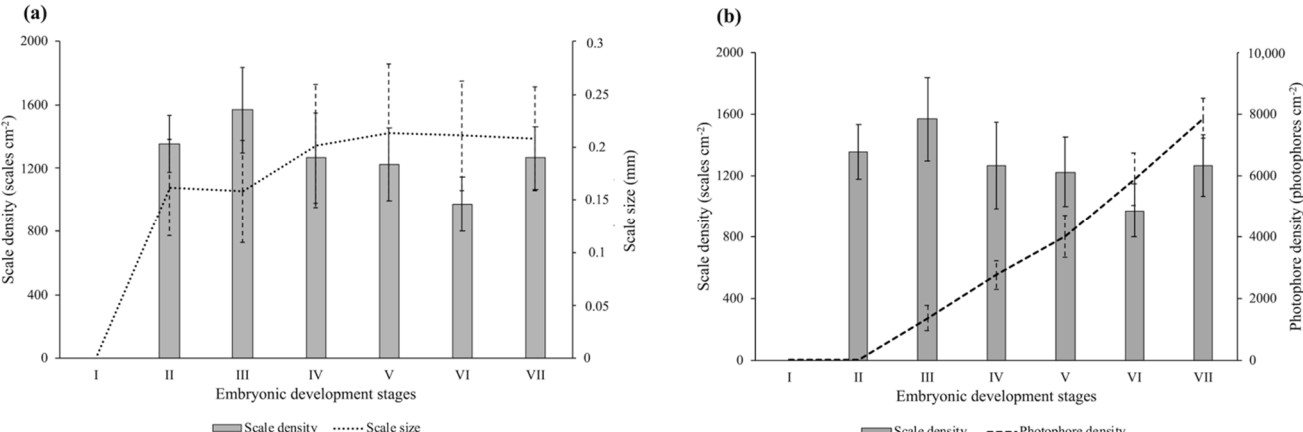

**Figure 3.** Scales' and photophores' parameters across embryogenesis for each analyzed stage. (**a**) Evolution of scales density and scales size. (**b**) Evolution of photophore density and scales density. Data are means ± standard deviation.

### 3.3. Encephalopsin Associated-Expression Pattern

In parallel to the development of photophores, encephalopsin expression was followed during embryonic light organ development (Figure 2d). The encephalopsin immunolabelling appears at stage IV, mainly at the top of the photophore in the ILS and lens cell membranes. Encephalopsin continues to be expressed in later developmental stages. Therefore, encephalopsin expression appears to be concomitant with the maturation of protophotophores, when light emission starts to occur. Immunodetection is never observed within the unique photocyte, while weaker labeling is detected at the photophore-surrounding epidermis for the adult stage. Moreover, the observed immunolabelling is consistent throughout the experimental repetition for each stage. Retina control experiments do not show any labeling, even with other visual opsin types (Supplemental Figure S1). Controls with the omission of the primary antibody, performed either on skin or retina sections, do not present any labeling (Supplemental Figure S1).

### 4. Discussion

In analyses of the *S. aliae*, skin parameters reveal that the average size of the scales, once they have appeared in the form of a fine, circular blank area, increases only slightly with the shark's growth. However, a relationship is present between scale size and density. Results show that when the size of the scales increases, the density tends to decrease, as space available for their development is more limited with their growth. The scale density remains relatively constant during embryonic development.

Little information is available on the scale development in luminous sharks, which does not allow a comparison between the different luminous shark families (i.e., Dalatiidae, Etmopteridae, and Somniosidae). Across the deep-sea species, dalatiid and etmopterid species have been shown to display a low placoid scale coverage (i.e., less than 40% of the skin) [37]. The scale shape of dalatiids (pavement-shaped) and etmopterids (pavement-, spine-, bristle-shaped) regarding scale densities and space between scales would then allow a higher luminescence propagation than other covering and tight scale shapes, such as shield-shaped scales [37,41]. The hypothesis of a space trade-off was suggested, reducing the putative defense or hydrodynamic scale function to allow a better light emission [19,37,41].

In contrast to the scales' density, the photophore density increases significantly in *S. aliae* during ontogenesis. These results contrast with those obtained by Claes and Mallefet in *E. spinax*, whereas the density of photophores tends to decrease during development as the diameter of the light organs increases [30]. This difference could be explained by the difference in the size of the light organs between Dalatiidae and Etmopteridae, with

dalatiid photophores being, on average, 50% smaller than those of etmopterids (maximum diameter of ~100 µm for Dalatiidae and ~200 µm for Etmopteridae [19]).

In addition, comparison of *S. aliae* and different adults etmopterid species photophores densities (*E. spinax* [42], *E. molleri* [43], and *E. splendidus* [44]) reveals a significantly higher density in *S. aliae* compared to Etmopteridae species (Table 1). The smalleye pygmy shark does not show various luminescent zones, with differences in photophore density depending on the body regions, conversely to etmopterids, which use bioluminescence for more complex purposes than just counterillumination [19]. In *S. aliae*, the photophore density is relatively uniform, suggesting a single use of luminescence as camouflage by counterillumination [30,32,41,45].

**Table 1.** TL interval and mean ventral photophore density of adult *S. aliae* and comparison with three Etmopteridae species.

| Species (Adult) | TL Interval (mm) | Mean $P_{De} \pm$ s.d. (Photophores cm$^{-2}$) | Replicates | Reference |
|---|---|---|---|---|
| *E. spinax* | 310–510 | $2883 \pm 232$ | n = 20 | [42] |
| *E. molleri* | 400–465 | $3862 \pm 193$ | n = 8 | [43] |
| *E. splendidus* | 203–235 | $4620 \pm 360$ | n = 3 | [44] |
| *S. aliae* | 143–225 | $7863 \pm 636$ | n = 16 | Present study |

TL, total length; $P_{De}$, photophores density. Data are mean $\pm$ s.d. (standard deviation).

Analysis of photophore development during Dalatiidae species embryogenesis reveals conserved morphogenesis across luminous sharks (Figure 4). The organogenesis of *S. aliae* photophores shows four similar developmental steps as in *E. spinax* [30,31], supporting the idea of a photophore stereotypical formation process among embryogenesis of luminous sharks (Figure 4). First, pigmented cells appear between the epidermis and the dermis. Then, pigmented cells will gather to form the pigment layer of the protophotophore with an ILS and differentiated cells which form the lens cells. The last step in the formation of the photophore is the appearance of a fluorescent vesicle within the protophotophore, which then becomes a functional photophore, capable of emitting light [30,31].

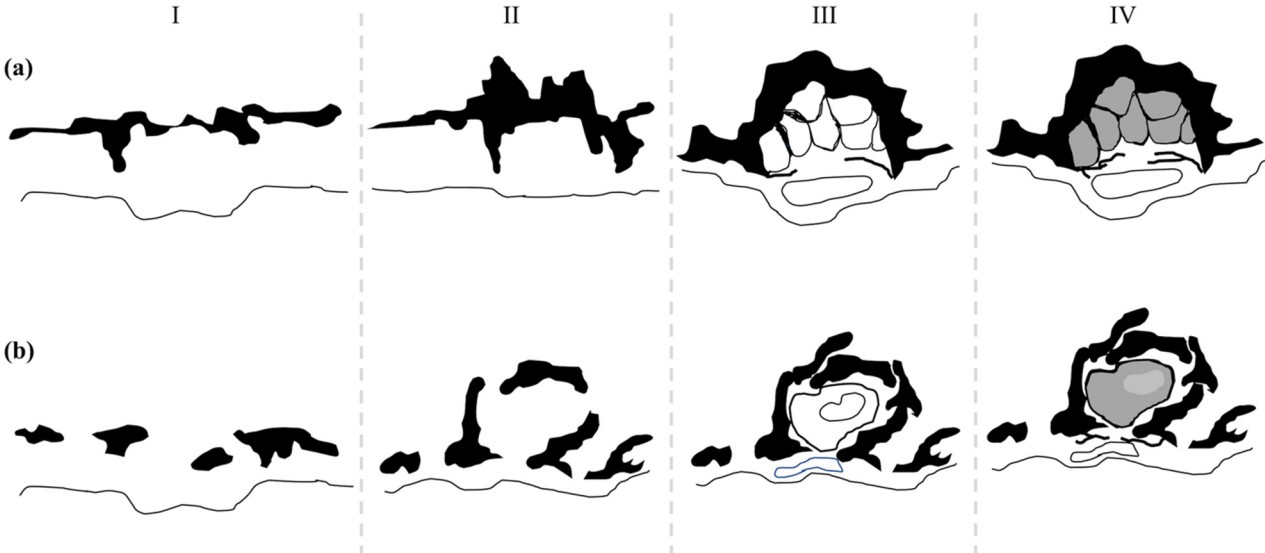

**Figure 4.** Schematic representation of (**a**) *Etmopteridae* and (**b**) *Dalatiidae* stereotypical photophores development steps. I: pigmented cells aggregation; II: cup-shaped pigmented cells formation; III: protophotophore (without autofluorescent vesicles), lens, and ILS cells appearance; IV: maturation of protophotophore to photophore with autofluorescent functional photocyte and bioluminescence capability acquisition.

Unlike etmopterid photophores, *S. aliae* photophores contain only one smaller photocyte within the luminescent organ [19,33]. In etmopterids, Claes and Mallefet [30] suggest that the acquisition of light compounds should be accomplished by maternal transfer via the yolk sac, as is the case for other luminescent fish species [46–48]. The same process of luminescent compound acquisition may be suggested in *S. aliae*; however, further studies should be conducted to confirm this hypothesis.

High homologies in photophore organogenesis between Dalatiidae and Etmopteridae support a common ancestral development, which is in accordance with the unique ancestral origin of sharks' luminescence [19]. The ability to emit light would have appeared only once in the luminous shark's evolution, allowing them to camouflage and escape predators. The rapid radiation and speciation of Etmopteridae, the most diverse and abundant family among the order Squaliformes, would have been accompanied by a photophores reorganization in different luminous patterns to allow, in addition to counterillumination, intraspecific communication and aposematic function in dark benthopelagic waters [19,49].

Photophores are fully formed and already capable of emitting light while the embryo is still in the uteri. This suggests that, as in *E. spinax* [30,31], juveniles can produce luminescence from birth and use it as camouflage by counterillumination. Results also highlight the co-expression of a photoreceptor, an encephalopsin, and the ability to produce light when photocyte maturation occurs. Encephalopsin expression, more intense in ILS cells, suggests that it could affect this structure, as demonstrated for *E. spinax* [25,31,50]. ILS is closely linked to light emission regulation, acting like a shutter allowing more or less light to pass through melanophore pigments' motion [19,25]. The presence, expression pattern of the encephalopsin, and the luminescence ability of the embryo within the uterus suggest a close relationship between extraocular perception and light emission by the *S. aliae* photophores. Furthermore, the overlap of the emission spectrum of *S. aliae* ($\lambda_{max}$: 457 nm [44]) and the encephalopsin absorbance spectrum ($\lambda_{max}$: 445 nm [25]) supports the opsin ability to perceive the shark own emitted light. Considering (i) the consistency of labeling during the experiments (at least ten sections for each embryonic stage and adults) and (ii) the control performed on *S. aliae* retina highlighting no cross labeling with closely related visual opsin sequences, our results are robust. Moreover, previous immunohistofluorescence and western blot results obtained with the same anti-encephalopsin antibody demonstrated a similar immunodetection for a phylogenetic closely related *Etmopterus* species, for which transcriptomic data reveal the expression of a highly homologous encephalopsin sequence [31,50]. All these results are consistent with the validity of our detection.

In *E. spinax*, light perception by encephalopsin results in a transduction cascade involving G protein, inositol triphosphate (IP$_3$), intracellular calcium storage concentration, Ca$^{2+}$-dependent calmodulin, calcineurin, and in fine cytoplasmic dynein-based cellular motor, which will allow moving the pigment granules from the periphery to the nuclei of the ILS melanophores, thus allowing more light to pass out of the photophore [25].

This pathway was shown to be coupled with pathways involving the light emission regulating hormones [25]. Thus, encephalopsin, in concert with hormones and neuromodulators, acting on pigment location modulation in the ILS and directly on photocyte light emission [18–20,25], affects the pigment motion in the ILS of photophores to control more precisely the light emission by the luminous organ.

Based on the *S. aliae* results, extraocular perception of emitted bioluminescence is strongly suggested for this species. The same phototransduction pathway may act as a feedback control mechanism in *S. aliae* allowing precise control of bioluminescence, concomitantly with hormones and neuromodulators, to perform an efficient counterillumination camouflage. Control of light emission by extraocular opsins, demonstrated for the lanternshark, has been suggested in other phylogenetically distinct bioluminescent species such as a ctenophore, an ophiuroid, and a deep-sea shrimp [21–24]. With the increasing number of species displaying evidence of a photoemission/photo-perception coupling at the light cell/organ level, the idea of a functional convergent evolution between the two

light-related processes is putatively concomitant as the bioluminescence systems' evolution emerges. Nevertheless, further studies are still needed to confirm this hypothesis.

**Supplementary Materials:** The following supporting information can be downloaded at: https://www.mdpi.com/article/10.3390/d14121100/s1, Figure S1: Immunohistological control. (a) Immunodetection test with anti-encephalopsin in *S. aliae* retina. (b) Immunodetection test with primary antibody omission in *S. aliae* retina. (c) Immunodetection test with primary antibody omission in photogenic ventral skin presenting photophore of *S. aliae*. All sections are illustrated in fluorescence and bright light microscopy. File S1: Statistical analyses on photophore density ($P_{De}$), scale density ($S_{De}$), and scale size ($S_{Si}$). (a) Means of TL, $P_{De}$, $S_{De}$, and $S_{Si}$ for every *S. aliae* development stage. (b) Pairwise comparisons using Dwass–Steele–Critchlow–Fligner all. Significative code: 0 '***'; 0.001 '**'; 0.01 '*'; 0.05 '.'; 0.1 ''; 1. (c) Dwass–Steele–Critchlow–Fligner all pairs test for photophores density, scales density, and scales size. (d) Linear relation between scales density and scales size.

**Author Contributions:** Conceptualization, L.D.; data curation, L.D.; formal analysis, L.D.; funding acquisition, J.M.; investigation, L.D.; methodology, L.D.; project administration, J.M.; resources, H.-C.H. and J.M.; software, L.D.; validation, L.D. and J.M.; writing—original draft, L.D.; writing—review and editing, L.D., H.-C.H. and J.M. All authors have read and agreed to the published version of the manuscript.

**Funding:** This research was funded by F.R.S.-FNRS, grant number T.0169.20 and a by travel grant to J.M. (grant number: 2018/6863).

**Institutional Review Board Statement:** Ethical review and approval were waived for this study since all the studied shark specimens were caught as by-catch.

**Data Availability Statement:** Data is contained within the article.

**Acknowledgments:** Authors thanks the anonymous fisherman from Donggang harbor for the help provided during field sessions. The authors also acknowledge Yannick Van den Bossche for his contribution to the study during his master thesis. This study is the contribution BRC 396 of the Biodiversity Research Center from the Earth and Life Institute—Biodiversity (UCLouvain—ELIV) and the "Centre Interuniversitaire de Biologie Marine" (CIBIM).

**Conflicts of Interest:** The authors declare no conflict of interest.

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
