# Peer review of "Photophore Morphogenesis and Extraocular Encephalopsin Expression during the Embryogenesis of Smalleye Pygmy Shark (Squaliolus aliae)"

_diversity, doi:10.3390/d14121100_

Round 1

Reviewer 1 Report

The authors are describing about observations of photophres during the timing of their formation by taking out embryos from gravid females of Squalious aliae. Also, the authors clarified the expression sites of encephalopsin during that timing by immunostaining using anti-human encepahlospsin antibody. This successfully observed the photophore formation at the embryo stage of Squalious aliae taking the advantage of ovoviviparous, as previously performed for Etmopterus spinax. This work expand our knowledge about luminous organ generation in shark by providing data adding to the previous known species, Etmopterus spinax.

The authors used anti-human encepalopsin antibody for immunostaining of Sqalious aliae sections. However, no amino acid sequence data is provided for the S. aliae encepahlopsin homolog. The authors showed negative control experiments, but specificity and reactivity are not clear. If authors provide western-blotting data using tissues around photophore, cross-reactivity to other proteins and molecular size of the target protein would be shown.

In Figure. 4, caption to IV is lacking.

Author Response

The authors are describing about observations of photophores during the timing of their formation by taking out embryos from gravid females of Squalious aliae. Also, the authors clarified the expression sites of encephalopsin during that timing by immunostaining using anti-human encephalopsin antibody. This successfully observed the photophore formation at the embryo stage of Squaliolus aliae taking the advantage of ovoviviparous, as previously performed for Etmopterus spinax. This work expands our knowledge about luminous organ generation in shark by providing data adding to the previous known species, Etmopterus spinax.

Thanks for this nice summary

The authors used anti-human encephalopsin antibody for immunostaining of Squaliolus aliae sections. However, no amino acid sequence data is provided for the S. aliae encephalopsin homolog. The authors showed negative control experiments, but specificity and reactivity are not clear. If authors provide western-blotting data using tissues around photophore, cross-reactivity to other proteins and molecular size of the target protein would be shown.

We understand the point, nevertheless, in view of the consistency of labeling during the experiments (at least 10 sections for each embryonic stages and female), the control performed on S. aliae retina (highlighting no cross labeling with closely related visual opsins sequences) and, the similar immunolabeling obtain for the phylogenetic closely related Etmopterus species (Delroisse et al., 2018; which validated the use of this antibody to label encephalopsin in at least Etmopterus spinax - IHF, WB, Sequence comparison), we are confident on our immunohistofluorescence results.

In Figure. 4, caption to IV is lacking.

Due to editing issue, you did not view the full version of the Figure 4 caption. It has been fixed now

Reviewer 2 Report

The manuscript described the photophore morphogenesis and extraocular opsin of pigmy shark depending on growth stage. It is clear results and discussion. The broad reader could be interested in this manuscript. I have some comments and suggestions as follows.

1)   In Introduction, the authors had better to introduce the lifestyle of pigmy shark; lifecycle (how long live?), food, predator, anti-predator etc.

2)   In Figure 2 (d), the faint red labeling is shown. The authors should explain this point. Is this protophotocyte? Not photocyte?

3)   If the authors show the immunehistological results, the authors should confirm the expression protein level using by Western blotting or immunoassay because the anti-human antibody must be checked.

4)   Figure 3 were not clear. The authors should change new figures.

5)   In the legend of Figure 4, the authors should explain the stage “IV”.

Author Response

The manuscript described the photophore morphogenesis and extraocular opsin of pigmy shark depending on growth stage. It is clear results and discussion. The broad reader could be interested in this manuscript. I have some comments and suggestions as follows.

Thanks for these nice comments

1)   In Introduction, the authors had better to introduce the lifestyle of pigmy shark; lifecycle (how long live?), food, predator, anti-predator etc.

Thanks for the suggestion. To date, the ecology of S. aliae remain virtually unknown excepted for partly described repartition and ovoviviparity. We have implemented few sentences at the end of the introduction. See lines 88-93

2)   In Figure 2 (d), the faint red labeling is shown. The authors should explain this point. Is this protophotocyte? Not photocyte?

The faint red labeling shown in Figure 2d is mentioned in line 230. Concerning the issue about protophotophore and photophore, you are right. Changes have been made accordingly in Figure 2 and text to underline the maturation of the light organ from embryonic stage 3. Thanks to have spotted this error.

3)   If the authors show the immunohistological results, the authors should confirm the expression protein level using by Western blotting or immunoassay because the anti-human antibody must be checked.

We understand the point, nevertheless, in view of the consistency of labeling during the experiments (at least 10 sections for each embryonic stages and females), the control performed on S. aliae retina (highlighting no cross labeling with closely related visual opsins sequences) and, the similar immunolabeling obtain for the phylogenetic closely related Etmopterus species (Delroisse et al., 2018; which validated the use of this antibody to label encephalopsin in at least Etmopterus spinax - IHF, WB, Sequence comparison), we are confident on our immunohistofluorescence results.

4)   Figure 3 were not clear. The authors should change new figures.

Figure 3 has been modified

5)   In the legend of Figure 4, the authors should explain the stage “IV”.

Due to editing issue, you did not view the full version of the Figure 4 caption. It has been fixed now

Reviewer 3 Report

This manuscript describes the initial development of photophores in a species of Dalatiid shark. This study also shows that an opsin called encaphalopsin is expressed in the photophore and is expressed in development when the photophore is assumed to be able to produce light. The authors draw upon the similarity of these results to those found in another bioluminescent shark in a different family (Etmopterus spinax; Etmopteridae) to argue for a shared origin of light organs (based on similar development and encephalopsin presence) that light sensing is important to light generation (based on the shared presence of encephalopsin, which presumable senses light).

I would recommend that you do away with all of your abbreviations for the terms that you measured (e.g. Pde, Sde, etc). These terms would work more effectively if they are simply written out (photophore density, scale density etc.). By abbreviating the terms you are giving the reader things to remember while you could otherwise just write out the terms that are both concise and clear.

I also think that you should consider adding more images into your paper – Figure 2 is nice, but it would also be nice to have some larger images as well (since the images in Figure 2 will be quite small).

And I also would suggest for you to add your statistical results onto the graphs you have made. My other comments are presented below in the order they appear in the manuscript.

Title: I think ‘pigmy’ should be pygmy.

Line 39: “such as cnidarian[ …]” should be something like “; for example, cnidarian, ctenophores, and most fishes control light production via adrenergic stimulation [5-10], and echinoderms control luminescence through a cholinergic control [11-13]”

Line 46-47: “Recently, an increasing 46 among of evidence of a duality between[…]” This phrase doesn’t make sense – please edit for grammar and clarity.

Line 67: Maybe explain why it is interesting that encephalopsin is expressed in the ILS membranes. I’m guessing it’s because it means that this tissue is also involved in light sensing (although perhaps I’m incorrect), but you should state that to make it obvious since this point is important for your paper!

Line 81: “encephalopsin-like are assumed” seems wrong – I think maybe you mean to say “are found” instead of “are assumed”? As far as I can tell, these are your main results and not things you are assuming to do your investigations, so you shouldn’t say that you are assuming both a pattern of similar morphogenesis as etmopterids and an association between light production and photo-perception through encephalopsin. Instead you should perhaps say that these are things that you found.

Lines 104-110: How are you measuring mean scale size? (is this the length of the crown, the area of the crown, some other measurement?) And how are you measuring skin pigmentation status? Photophore and placoid scale density seem obvious measurements and you give some explanation of them (which is good!), but mean scale size and skin pigmentation status are not as obvious and need to be explained.

Figure 3 caption: for part a) is Scales relative size the same as mean scale size?

Figure 3: Why display scale density as a barchart in part a) and then as a mean/sd point in part b)? The same data should likely be displayed the same way. Also, why skin stage 1 in part a) but include it in b)?

215-216: I know you cite a few papers here but it would be good to briefly explain why the scale shapes in these families of sharks better allows a ‘higher light transmission efficiency’.

Line 229: Did you actually measure photophore density on different body regions? How do you know there aren’t differences in photophore density across body regions?

Lines 231-233: This sentence is hard to understand – maybe consider revising to make it clearer for readers what you mean here? Not sure what you mean by increasing centripetally, or median area of the ventral side, or a unique use of luminescence as camouflage by counterillumination (most luminescence is used for counterillumination in the deep ocean, so why is this case “unique”?)

Lines 250-258: Somewhere in here you should explain the phylogenetic position of the three bioluminescent shark families – are they all three monophyletic with one another? Which would further support the appearance of just a single instance of photophore evolution? I would also suggest adding some explanation of the phylogenetic placement of these three families into the introduction and perhaps also a figure to show why your work here is valuable – you are able to confirm what we know already from Etmopteridae in a different shark family…but readers will want to know how closely related these families are.

Author Response

This manuscript describes the initial development of photophores in a species of Dalatiid shark. This study also shows that an opsin called encephalopsin is expressed in the photophore and is expressed in development when the photophore is assumed to be able to produce light. The authors draw upon the similarity of these results to those found in another bioluminescent shark in a different family (Etmopterus spinax; Etmopteridae) to argue for a shared origin of light organs (based on similar development and encephalopsin presence) that light sensing is important to light generation (based on the shared presence of encephalopsin, which presumable senses light).

I would recommend that you do away with all of your abbreviations for the terms that you measured (e.g. Pde, Sde, etc). These terms would work more effectively if they are simply written out (photophore density, scale density, etc.). By abbreviating the terms you are giving the reader things to remember while you could otherwise just write out the terms that are both concise and clear.

Thanks for this comment. Abbreviations have been deleted and replaced by the full-length terminology all along the manuscript.

I also think that you should consider adding more images into your paper – Figure 2 is nice, but it would also be nice to have some larger images as well (since the images in Figure 2 will be quite small).

We think we have placed figures/images essential to understanding the manuscript and illustrating our results. Image 2 will be a full-page image.

And I also would suggest for you to add your statistical results onto the graphs you have made. My other comments are presented below in the order they appear in the manuscript.

Thanks for the suggestion, nevertheless, Figure 3 is already well charged, and all the statistical data are present in supplementary table 1.

Title: I think ‘pigmy’ should be pygmy.

Indeed, change has been made

Line 39: “such as cnidarian[ …]” should be something like “; for example, cnidarian, ctenophores, and most fishes control light production via adrenergic stimulation [5-10], and echinoderms control luminescence through a cholinergic control [11-13]”

Done

Line 46-47: “Recently, an increasing among of evidence of a duality between[…]” This phrase doesn’t make sense – please edit for grammar and clarity.

Thanks for the comment, the sentence has been rewritten to better understandability.

Line 67: Maybe explain why it is interesting that encephalopsin is expressed in the ILS membranes. I’m guessing it’s because it means that this tissue is also involved in light sensing (although perhaps I’m incorrect), but you should state that to make it obvious since this point is important for your paper!

Yes, you are right. Early detection of encephalopsin in ILS during photophore morphogenesis clearly suggests a bioluminescence photo-perception capability.

Line 81: “encephalopsin-like are assumed” seems wrong – I think maybe you mean to say “are found” instead of “are assumed”? As far as I can tell, these are your main results and not things you are assuming to do your investigations, so you shouldn’t say that you are assuming both a pattern of similar morphogenesis as etmopterids and an association between light production and photo-perception through encephalopsin. Instead you should perhaps say that these are things that you found.

Indeed, we must go to the point. We present results that we FOUND. the sentence has been modified accordingly.

Lines 104-110: How are you measuring mean scale size? (is this the length of the crown, the area of the crown, some other measurement?) And how are you measuring skin pigmentation status? Photophore and placoid scale density seem obvious measurements and you give some explanation of them (which is good!), but mean scale size and skin pigmentation status are not as obvious and need to be explained.

Corresponding Materials and methods section has been modified to add methodological issue.

Figure 3 caption: for part a) is Scales relative size the same as mean scale size?

Yes, changes have been made in the Figure 3 caption.

Figure 3: Why display scale density as a barchart in part a) and then as a mean/sd point in part b)? The same data should likely be displayed the same way. Also, why skin stage 1 in part a) but include it in b)?

Figure 3 has been changed and scale density data were represented as bar chart in both graphs. Skin data for the stage 1 have also been implemented even if values are null.

215-216: I know you cite a few papers here, but it would be good to briefly explain why the scale shapes in these families of sharks better allows a ‘higher light transmission efficiency’.

Thanks for the comment. The sentence was modified to better highlight the trade-off hypothesis underlying a better luminescence propagation for these luminous shark placoid scale types

Line 229: Did you actually measure photophore density on different body regions? How do you know there aren’t differences in photophore density across body regions?

As mentioned in Materials and Methods section 2.2, only ventral skin patches (main area of light emission) have been analyses in this study.

Lines 231-233: This sentence is hard to understand – maybe consider revising to make it clearer for readers what you mean here? Not sure what you mean by increasing centripetally, or median area of the ventral side, or a unique use of luminescence as camouflage by counterillumination (most luminescence is used for counterillumination in the deep ocean, so why is this case “unique”?)

Thanks for the remarks. Changes have been made to clarify the meaning of the sentence.

Lines 250-258: Somewhere in here you should explain the phylogenetic position of the three bioluminescent shark families – are they all three monophyletic with one another? Which would further support the appearance of just a single instance of photophore evolution? I would also suggest adding some explanation of the phylogenetic placement of these three families into the introduction and perhaps also a figure to show why your work here is valuable – you are able to confirm what we know already from Etmopteridae in a different shark family…but readers will want to know how closely related these families are.

Thanks for the question and suggestions. A sentence stating the phylogenetic position of the three luminous shark families has been implemented in the introduction section. Besides, the discussion has been modified according to your suggestion. Concerning the addition of a new phylogenetic picture, we refer to a recently published phylogenetic representation of the Squaliformes where bioluminescence origin in shark is clearly indicated. See: https://doi.org/10.3390/oceans2040047